Lewin *et al. Implementation Science* 2018, **13**(Suppl 1):10

Implementation Science

**METHOD**　　　　　　　　　　　　　　　　　　　　　　　**Open Access**

# Applying GRADE-CERQual to qualitative evidence synthesis findings—paper 2: how to make an overall CERQual assessment of confidence and create a Summary of Qualitative Findings table

Simon Lewin[1,2*], Meghan Bohren[3], Arash Rashidian[4,5], Heather Munthe-Kaas[1], Claire Glenton[1], Christopher J. Colvin[6], Ruth Garside[7], Jane Noyes[8], Andrew Booth[9], Özge Tunçalp[3], Megan Wainwright[6], Signe Flottorp[1], Joseph D. Tucker[10] and Benedicte Carlsen[11]

## Abstract

**Background:** The GRADE-CERQual (Confidence in Evidence from Reviews of Qualitative research) approach has been developed by the GRADE (Grading of Recommendations Assessment, Development and Evaluation) Working Group. The approach has been developed to support the use of findings from qualitative evidence syntheses in decision making, including guideline development and policy formulation.

CERQual includes four components for assessing how much confidence to place in findings from reviews of qualitative research (also referred to as qualitative evidence syntheses): (1) methodological limitations, (2) coherence, (3) adequacy of data and (4) relevance. This paper is part of a series providing guidance on how to apply CERQual and focuses on making an overall assessment of confidence in a review finding and creating a CERQual Evidence Profile and a CERQual Summary of Qualitative Findings table.

**Methods:** We developed this guidance by examining the methods used by other GRADE approaches, gathering feedback from relevant research communities and developing consensus through project group meetings. We then piloted the guidance on several qualitative evidence syntheses before agreeing on the approach.

**Results:** Confidence in the evidence is an assessment of the extent to which a review finding is a reasonable representation of the phenomenon of interest. Creating a summary of each review finding and deciding whether or not CERQual should be used are important steps prior to assessing confidence. Confidence should be assessed for each review finding individually, based on the judgements made for each of the four CERQual components. Four levels are used to describe the overall assessment of confidence: high, moderate, low or very low. The overall CERQual assessment for each review finding should be explained in a CERQual Evidence Profile and Summary of Qualitative Findings table.

\* Correspondence: simon.lewin@fhi.no
[1]Norwegian Institute of Public Health, Oslo, Norway
[2]Health Systems Research Unit, South African Medical Research Council,
Cape Town, South Africa
Full list of author information is available at the end of the article

**Conclusions:** Structuring and summarising review findings, assessing confidence in those findings using CERQual and creating a CERQual Evidence Profile and Summary of Qualitative Findings table should be essential components of undertaking qualitative evidence syntheses. This paper describes the end point of a CERQual assessment and should be read in conjunction with the other papers in the series that provide information on assessing individual CERQual components.

**Keywords:** Qualitative research, Qualitative evidence synthesis, Systematic review methodology, Research design, Methodology, Confidence, Guidance, Evidence-based practice, GRADE

## Background

GRADE-CERQual (Confidence in the Evidence from Reviews of Qualitative Research) is an approach for assessing how much confidence to place in review findings from systematic reviews of qualitative research or qualitative evidence syntheses (QES). The approach is being developed by the GRADE-CERQual Project Group, a subgroup of the GRADE (Grading of Recommendations Assessment, Development and Evaluation) Working Group. The importance of assessing confidence in qualitative evidence is discussed in the first paper in this series [1].

The GRADE-CERQual approach (hereafter referred to as CERQual) is applied to individual review findings from a qualitative evidence synthesis [1]. We define a review finding as an analytic output (e.g. a theme, category, thematic framework, theory or contribution to theory) from a qualitative evidence synthesis that, based on data from primary studies, describes a phenomenon or an aspect of a phenomenon. By 'phenomenon', we mean the issue that is the focus of the qualitative inquiry—that is, 'what we want our research to understand…[]… explain, or describe' [2] (p129). How review findings are defined and presented depends on factors such as the review question, the synthesis methods utilised and the intended purpose or audience of the synthesis [3]. For example, the aims of different approaches to QES range from identifying and describing key themes, to seeking more generalizable or interpretive explanations that can be used for building theory [3, 4].

GRADE-CERQual includes four components for assessing how much confidence to place in review findings: (1) the methodological limitations of the individual qualitative studies contributing to a review finding, (2) the coherence of the review finding, (3) the adequacy of data supporting a review finding and (4) the relevance of the data from the primary studies supporting a review finding to the context (perspective or population, phenomenon of interest and/or setting) specified in the review question. Making an overall assessment of confidence in a review finding involves moving from the judgements made for each CERQual component to a final assessment. The overall assessment of confidence in a review finding takes into

account the concerns identified in relation to each of the four components and, in our experience, is an iterative process that benefits from discussion among the review team or those making the CERQual assessment (when this is being done for an existing synthesis).

## Aim

The aim of this paper, the second in this series (Fig. 1), is to describe and discuss the process for making an overall assessment of confidence in a review finding and to outline how to create a CERQual Evidence Profile and a Summary of Qualitative Findings table. A key part of communicating an overall CERQual assessment is providing an explanation for the confidence assessment, based on the concerns identified in relation to each component. The review team therefore needs to create a Summary of Qualitative Findings table to display a summary of each review finding, the CERQual assessment of confidence in that finding and the explanation for that assessment.

In this paper, we discuss the following processes: firstly, moving from a review finding to a summary of a review finding; secondly, determining the review findings to which CERQual should be applied; thirdly, making an overall CERQual assessment of confidence in each individual review finding; and, finally, creating a CERQual Evidence Profile and a Summary of Qualitative Findings (SoQF) table. Papers 3, 4, 5 and 6 in the series describe how to assess each CERQual component (i.e. methodological limitations [5], coherence [6], adequacy [7], and relevance [8]). These component papers are closely related to this paper on making an overall CERQual assessment of confidence and creating a Summary of Qualitative Findings table. We have placed this paper before the four CERQual component papers as we think that it will be helpful for readers to understand how the component assessments will be used before discussing the details of how to apply each component. Key definitions are provided in Additional file 1.

## How CERQual was developed

The initial stages of the process for developing CERQual, which started in 2010, are outlined elsewhere [3]. Since

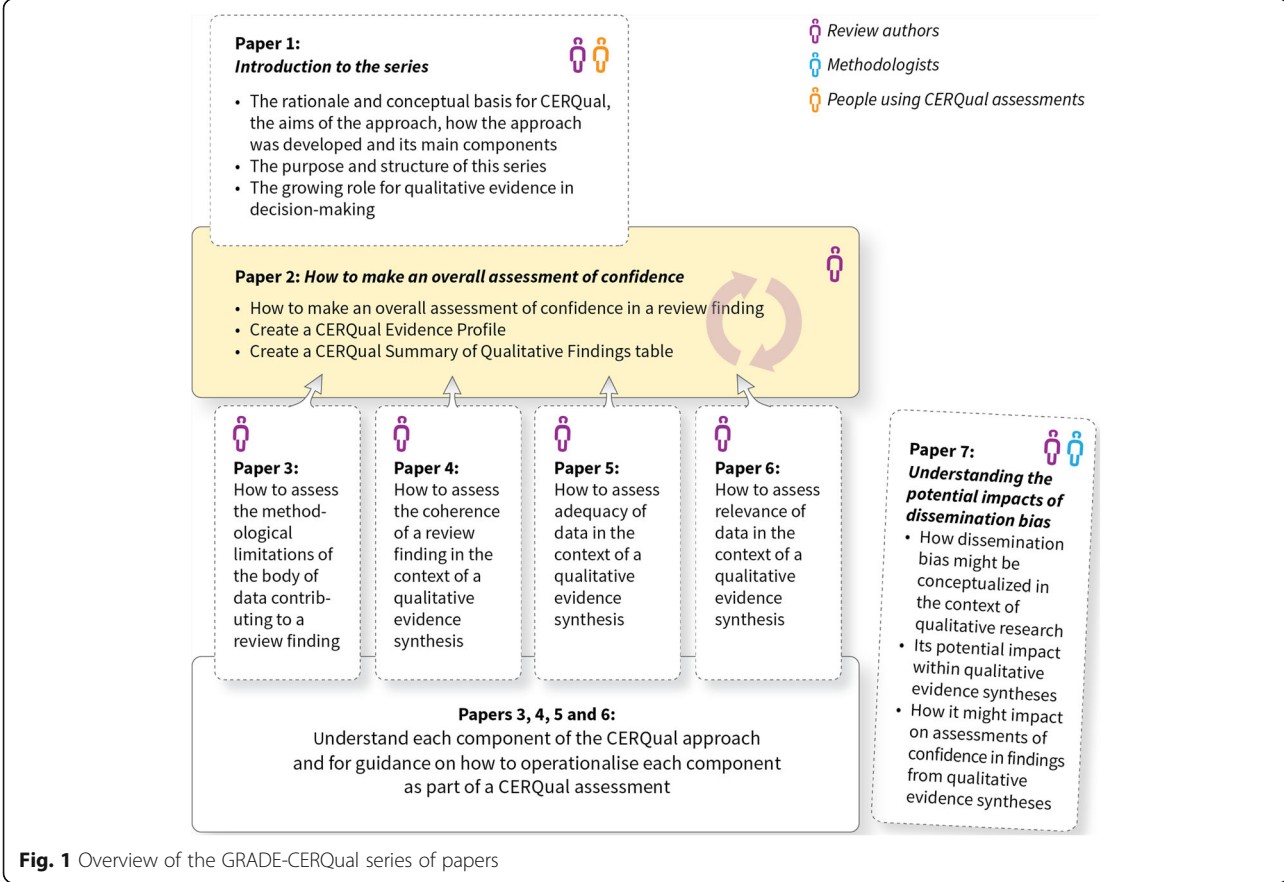

**Fig. 1** Overview of the GRADE-CERQual series of papers

then, we have used a number of methods to further refine the definitions of each component and the principles for application of the overall approach. We used a pragmatic approach to develop the methods for making an overall CERQual assessment of confidence and creating a Summary of Qualitative Findings table. We examined the methods used by other GRADE approaches, talked to experts in the field of qualitative evidence synthesis, developed consensus through multiple face-to-face CERQual project group meetings and teleconferences, and gained feedback from ongoing engagement with the qualitative evidence synthesis community. We presented an early version of the CERQual approach in 2015 to a group of methodologists, researchers and end users with experience in qualitative research, GRADE or guideline development. We further refined the approach through training workshops, seminars and presentations during which we actively sought, collated and shared feedback; by facilitating discussions of individual CERQual components within relevant organisations; through applying the approach within diverse qualitative evidence syntheses [9–19]; and through supporting other teams in using CERQual [20, 21]. As far as possible, we used a consensus approach in these processes. We also gathered feedback from CERQual users through an

online feedback form and through short individual discussions with members of the review teams. The methods used to develop CERQual are described in more detail in the first paper in this series [1].

## Making an overall CERQual assessment of confidence and creating a Summary of Qualitative Findings table
### Moving from a review finding to a summary of a review finding

Once the review findings for a qualitative evidence synthesis have been formulated, members of the review team may find it helpful to begin drafting short statements or 'summaries of findings' that provide a short but clear description of each review finding. This is especially important when the texts of review findings are lengthy, as often is the case in qualitative evidence syntheses. Summaries of review findings can have different benefits. First, they may help the review team to identify the central idea of each finding, including any explanatory aspects. Second, the process of developing these summaries may also promote an iterative and reflexive discussion within the review team regarding the key content of each review finding. Third, the summaries of findings are the starting point for creating a CERQual Evidence Profile and Summary of Qualitative Findings

(SoQF) table, which present each review finding along with its CERQual assessment. Fourth, the summaries of review findings in the CERQual Evidence Profiles and SoQF may be used in guideline and other decision making processes to clearly and concisely present the evidence, and confidence in this evidence, to decision makers and other end users. A summary of a review finding therefore needs to be written with the end users and key stakeholders for the review in mind—the review team should consider what these potential users would want to know. Review teams also need to strike a balance between splitting issues emerging from the synthesis into many review findings, resulting in findings that lose their usefulness to end users and their ability to reasonably represent the phenomenon of interest, and creating a smaller number of broad findings that may oversimplify and fail to adequately capture variations across different contexts. Table 1 provides guidance on writing a summary of a review finding for an Evidence Profile and SoQF table. Table 2 provides examples summaries of review findings, including ways in which the writing of these findings might be improved.

### Determining the review findings to which CERQual should be applied

Our intention is that CERQual can be applied to the full range of types and levels of review findings. For example, a qualitative evidence synthesis may include review findings ranging from more descriptive or aggregative to more interpretive or configuring [4], as well as from more narrow (for example, in relation to a specific health care setting) to more broad (for example, cutting across several different kinds of social care settings). In general, the review team would assess all review findings emerging from a QES, but there may be circumstances in which this is not feasible or appropriate. For instance, some of the findings from a QES may be particularly relevant to a decision making process and the review team may therefore choose to apply CERQual to those findings only.

There are important reasons why it may be useful to apply CERQual to as many of the review findings as possible. Firstly, it is easier to undertake a CERQual assessment as part of the process of developing review findings, when the necessary data are easily accessible. Secondly, it is not always possible to predict which findings may be useful for a decision. Applying CERQual to all review findings means that individual review findings can be more quickly integrated into decision making processes. Where the review team decides not to apply CERQual to all review findings, a justification for this should be provided.

For a particular synthesis, the review team also needs to decide at which level of interpretation to apply CERQual. For instance, users may want to apply CERQual to a broad theory emerging from a review which, in turn, may draw on several individual review findings. In other cases, users may want to apply CERQual to a set of descriptive review findings (for example, experiences of an intervention among different population groups). To date, the experience in applying CERQual to more explanatory or interpretive review findings is limited—Additional file 2 provides further guidance on this.

### Making an overall CERQual assessment of confidence in each individual review finding

Within the CERQual approach, confidence in the evidence is an assessment of the extent to which a review finding is a reasonable representation of the phenomenon of interest. Confidence should be assessed for each review finding individually, and not for the synthesis as a whole. An assessment of our confidence in a review finding is based on the judgements made for each of the four CERQual

**Table 1** Guidance on writing and structuring a summary of a review finding for a CERQual Evidence Profile and Summary of Qualitative Findings (SoQF) table

**How much detail should a summary of a review finding include?**
- You should make the summary as explicit as possible: for example, a hypothesized connection or pathway between factors should not be implied but should be described clearly
- The level of detail that you include in a summary of a review finding will vary according to the nature of the finding and the style agreed by the review team:
  o Providing detailed contextual information in a summary of a review finding (e.g., 'In three studies from Japan, the UK and Ghana, young women reported that…') may give the impression that the finding does not represent a broader phenomenon and is not transferable (see [8] and Table 3). It may be more appropriate to include this descriptive information in the column of the SoQF table that lists the studies contributing to the review finding
  o You should only include geographic or other specifiers in a summary of a review finding if these form part of the explanation of the finding; are needed to understand the finding or the group or context to which the finding relates (for example, 'Health care providers reported that…' or 'Young men from rural areas experienced….'); or are critical to addressing the review question [8].
- Where a review is commissioned in support of a guideline, you may want to write the SoQF table to take into account the information needed to make a recommendation. For example, the summaries of review findings included in the SoQF table may focus on issues related to the acceptability and feasibility of the interventions being considered in the guideline and any important implementation considerations [25]

**How should the summaries of review findings be ordered and presented?**
- You can use titles and subheadings in the SoQF table to flag larger groupings of review findings
- You can use a broader introductory or summary sentence, or a one line header, at the start of a summary of a review finding to help users quickly understand the gist of the finding

**Table 2** Examples of summaries of review findings, including how these might be improved

| Example | Original summary of a review finding | Explanation of concerns with how the review finding has been written | Improved summary of a review finding |
|---|---|---|---|
| 1 | Lay health workers in child health studies in Angola, Zambia and Zimbabwe, and supervisors from these studies, as well as two studies in HIV/AIDS clinics in South Africa, expressed concern about the lay health workers' workload and the distances they had to cover. Lay health workers in Angola, South Africa, Zambia and Zimbabwe sometimes found it difficult to carry out all of their tasks because of this. | *Unnecessary contextual information included in the summary of a review finding* | Lay health workers and supervisors expressed concern about the lay health workers' workload and the distances they had to cover, and lay health workers sometimes found it difficult to carry out all of their tasks because of this. |
| 2 | Some parents wanted less information about childhood vaccination while other parents wanted a larger amount of information and/or information at a larger number of timepoints. Acceptance of vaccination varied among these parents. | *The summary review of a finding is not as explicit as possible—the hypothesised connection between the amount of information that parents would like to receive and their acceptance of vaccination is implied, but not spelled out* | The amount of information that parents would like to receive about childhood vaccination is linked to their acceptance of vaccination. Parents who had accepted vaccination as necessary typically wanted less information. Parents who questioned or refused vaccinations typically wanted a larger amount of information and / or information at a larger number of timepoints. |
| 3 | Women reported being beaten by health workers. | *This summary of a review finding provides insufficient detail to understand the situation* | Women reported experiencing physical force by health providers during childbirth, In some cases, women reported specific acts of violence committed against them during childbirth, but women often referred to these experiences in a general sense and alluded to beatings, aggression, physical abuse, a rough touch, and use of extreme force. Pinching, hitting and slapping (either with an open hand or an instrument) were the most commonly reported specific acts of violence. |

Adapted from [9, 12, 14]. Please note that an assessment of whether a summary of a review finding is sufficiently clear and explicit involves a judgement

components [5–8] (Fig. 2). Each CERQual component should initially be assessed individually. For each component, the assessment should be described using one of the following levels of concern:

- No or very minor concerns regarding [methodological limitations/coherence/adequacy/relevance] that are unlikely to reduce confidence in the review finding
- Minor concerns regarding [methodological limitations/coherence/adequacy/relevance] that may reduce confidence in the review finding [concerns to be described]
- Moderate concerns regarding [methodological limitations/coherence/adequacy/relevance] that will probably reduce confidence in the review finding [concerns to be described]
- Serious concerns regarding [methodological limitations/coherence/adequacy/relevance] that are very likely to reduce confidence in the review finding [concerns to be described]

Few review findings are likely to have no concerns regarding methodological limitations, coherence, adequacy

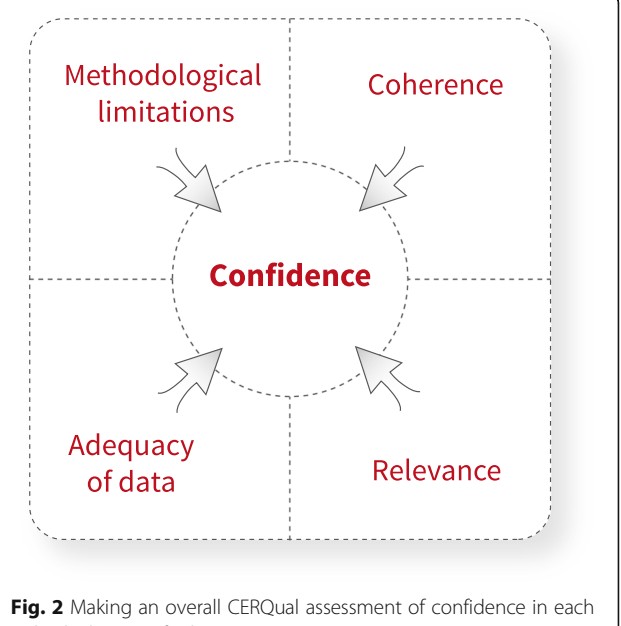

**Fig. 2** Making an overall CERQual assessment of confidence in each individual review finding

or relevance. However, in making judgements in relation to each of the CERQual components, the review team should be looking for *important* concerns that may impact on confidence in the evidence—very minor concerns need not be listed. Tables 3 and 4 provide examples of what constitutes different levels of concerns in relation to the four CERQual components, and further examples are provided in other papers in this series [5–8].

We do not recommend attempting to numerically score assessments for each component as this may give a false sense of precision regarding these assessments. CERQual assessments are judgements and our aim is to make these judgements as explicit and transparent as possible through providing explanations of the reasons for the assessments made. Any concerns with regard to each component should be described in the CERQual Evidence Profile in sufficient detail to allow users of the review findings to understand the reasons for the assessments made. Tables 3 and 4 provide examples of CERQual Evidence Profiles, including assessments for each component.

Four levels are used to describe the overall assessment of confidence in a review finding—high, moderate, low or very low—and these are defined in Table 3 in the first paper in this series [1]. All review findings start off by default as 'high confidence' and are then 'rated down' by one or more levels (for example, from high to moderate confidence) if there are concerns regarding any of the CERQual components (Table 5). This starting point of 'high confidence' reflects a view that each review finding should be seen as a reasonable representation of the phenomenon of interest unless there are factors that would weaken this assumption.

**Who should undertake a CERQual assessment?** CERQual should ideally be applied by the review team to their own review findings, given that prior familiarity with the evidence is needed in order to make reasonable judgments concerning each CERQual component. This has additional benefits for the qualitative evidence synthesis as it provides opportunities for reflection on the way in which the synthesis has been conducted and the review findings formulated.

We recommend that CERQual assessments for each review finding should be made through discussion between at least two members of the review team. This is preferable to use by a single review author as it offers an opportunity to discuss judgments within the review team, may assist members of the review team in clearly describing the rationale behind each assessment and can be a helpful part of the iterative and reflexive process of formulating review findings. Because CERQual assessments are judgements, there is likely to be variation across assessors. In such cases, a strength of CERQual is that it provides a system that guides assessors through the key components of this assessment and an approach to reporting that promotes transparency and an explicit record of the judgements involved [22]. We therefore anticipate that the CERQual approach will improve reliability in comparison to intuitive judgments [23].

While the focus of this paper is on the application of CERQual by the review team, a CERQual assessment can, in principle, be applied to review findings from well-conducted qualitative evidence syntheses undertaken by others. In this case, sufficient time should be devoted to gaining a full understanding of how the synthesis has been conducted and presented before applying CERQual. Initial guidance for applying CERQual to a synthesis done by another team is provided in Additional file 3. Our limited experience to date suggests that applying CERQual to a synthesis done by another team is likely to be a challenging and time-consuming process, in part, because creating a Summary of Qualitative Findings is an interpretive process.

**Practical considerations when undertaking a CERQual assessment** This paper provides broad guidance on how to approach a CERQual assessment. Detailed guidance on factors that may lead the review team to have concerns about a particular component is provided in the papers discussing each component [5–8].

An overall assessment of confidence is based on the assessments for each CERQual component. Currently, CERQual gives equal weight to each component because there is no evidence to suggest that the components should not be given equal importance. Further research is needed on whether giving different weights to different components would be more appropriate.

The review team should look iteratively across the four CERQual components in order to make an overall CERQual assessment. Review authors should also note potential interactions and overlaps between the components and avoid downgrading a review finding twice for the same concern across components [3]. For example, an insufficient quantity of data in studies contributing to a review finding might affect assessments of both coherence of the finding and adequacy of data. 'Double downgrading' may also be a concern when review findings are used in a decision making process. For example, users of CERQual assessments may have concerns about the relevance of the review finding to their decision question but may fail to notice that these concerns about relevance have already been taken into account in the overall CERQual assessment. Users should therefore be careful to check the reasons why confidence in a review finding has been graded down before making any

**Table 3** CERQual Evidence Profile—Example A

| Summary of review finding | Studies contributing to the review finding | Methodological limitations | Coherence | Adequacy | Relevance[a] | CERQual assessment of confidence in the evidence | Explanation of CERQual assessment |
|---|---|---|---|---|---|---|---|
| 1. Use of force: Women across the world reported experiencing physical force by health providers during childbirth. In some cases, women reported specific acts of violence committed against them during childbirth, but women often referred to these experiences in a general sense and alluded to beatings, aggression, physical abuse, a rough touch and use of extreme force. Pinching, hitting and slapping, either with an open hand or an instrument were the most commonly reported specific acts of physical violence. | 6, 9, 10, 13, 21, 61, 67, 68, 73, 75, 77, 80, 84, 86, 87, 91, 96, 97 | Moderate methodological limitations (6 studies with minor, 6 studies with moderate (unclear recruitment and sampling), and 3 studies with serious methodological limitations (unclear reflexivity, insufficiently rigorous data analysis) | No or very minor concerns about coherence | No or very minor concerns about adequacy | Minor concerns about relevance (5 studies with direct relevance, 8 studies with partial relevance, and 1 study with unclear relevance. 15 studies total from 10 countries, including 1 high income, 2 middle income and 7 low income countries. Geographical spread: 2 studies in Asia, 1 study in Europe, 1 study in LAC, 1 study in MENA, 1 study in South America, and 8 studies from sub-Saharan Africa.) | High confidence | 15 studies with moderate methodological limitations. Data from 10 countries across all geographical regions, but predominantly sub-Saharan Africa. No or very minor concerns about coherence and adequacy. |
| 2. Physical restraint: Women in Tanzania and Brazil reported physical restraint during childbirth through the use of bed restraints and mouth gags. | 86, 97 | Moderate methodological limitations (1 study with minor and 1 study with serious methodological limitations (inappropriate research design, no reflexivity, unclear ethical considerations)) | Minor concerns about coherence (Some concerns about the fit between the data from primary studies and the review finding) | Serious concerns about adequacy (2 studies in total with limited, thin data) | Moderate concerns about relevance (2 studies with partial relevance from 2 countries) | Very low confidence | Two studies (Tanzania and Brazil) with moderate methodological limitations. Limited, thin data from 2 countries. Minor concerns about coherence but limited data available. |

Review findings taken from [12] and adapted to fit the context of this article. The review findings presented here are drawn from the wider thematic synthesis undertaken for this review. The themes identified were summarised into evidence statements, as illustrated in this table. The methods are described in more detail in [12]

[a]When describing relevance judgements, consider the following prompts to help elucidate 'partial' and/or 'unclear' relevance: phenomenon of interest, population (including subgroups), setting, place, intervention, findings

**Table 4** CERQual Qualitative Evidence Profile—Example B

| Summary of review finding | Studies contributing to the review finding | Methodological limitations | Coherence | Adequacy | Relevance[a] | CERQual assessment of confidence in the evidence | Explanation of CERQual assessment |
|---|---|---|---|---|---|---|---|
| 1. While regular salaries were not part of many programmes, other monetary and non-monetary incentives, including payment to cover out-of-pocket expenses and "work tools" such as bicycles, uniforms or identity badges, were greatly appreciated by lay health workers. | 2, 5, 11, 12, 22, 29 | Minor methodological limitations (five studies with minor and one study with moderate methodological limitations (unclear recruitment and sampling strategy, no reflexivity)) | Minor concerns about coherence (some concerns about the fit between the data from primary studies and the review finding) | Minor concerns about adequacy (six studies that together offered moderately rich data) | Minor concerns about relevance (studies of lay health worker programmes from three continents and including a fairly wide range of different clients and health issues) | Moderate confidence | Minor concerns regarding methodological limitations, relevance, coherence and adequacy. |
| 2. Some unsalaried lay health workers expressed a strong wish for regular payment. | 5, 13 | Minor methodological limitations (both studies had minor methodological limitations (unclear sampling strategy, no reflexivity)) | Minor concerns about coherence (some concerns about the fit between the data from primary studies and the review finding) | Serious concerns about adequacy (only two studies, both offering thin data) | Moderate concerns about relevance (partial relevance as the studies were from only two settings, both of which were in Africa) | Low confidence | Moderate concerns regarding relevance and serious concerns regarding adequacy of data. |
| 3. Lay health workers, particularly those working in urban settings, reported difficulties maintaining personal safety when working in dangerous settings or at night. | 3, 15, 16, 25, 31 | Moderate methodological limitations (2 studies with minor methodological limitations, 2 studies with moderate methodological limitations (unclear ethical considerations and unclear statement of research aims) | Minor concerns about coherence (some concerns about the fit between the data from primary studies and the review finding) | Moderate concerns about adequacy (studies offered very thin data) | Minor concerns about relevance (studies of lay health worker programmes across three continents but for a limited range of health issues) | Moderate confidence | Moderate methodological limitations and moderate concerns regarding adequacy of data. |

Review findings taken from [14] and adapted to fit the context of this article. The review findings presented here are drawn from the wider thematic synthesis undertaken for this review. The themes identified were summarised into summaries of review findings, as illustrated in this table. The methods are described in more detail in [14]

[a]When describing relevance judgements, consider the following prompts to help elucidate 'partial' and/or 'unclear' relevance: phenomenon of interest, population (including subgroups), setting, place, intervention, findings

**Table 5** Practical guidance on making an overall CERQual assessment for a review finding

- Each overall CERQual assessment should ideally be made through discussion among the review authors. This process may also involve consulting with an expert group for a synthesis
- Using the CERQual Evidence Profile, look across the assessments you have made for each CERQual component. Note particularly any components for which you have serious concerns
- Decide whether you will 'rate down' (i.e., lower the level of confidence in the review finding) at all for the concerns identified and, if so, whether you will rate down by one or two levels. When making this overall assessment, consider the following:
  - Typically, the overall assessment of confidence should be rated down by at least one level for each component for which you have identified serious concerns
  - Where concerns in relation to a component are minor or moderate, it may not be necessary to rate down. However, if there are a number of such concerns, it may be appropriate to rate down by one level to represent two or more of these concerns
- When making a judgement on whether to 'rate down', also consider the following:
  - To some extent, the importance of concerns regarding a CERQual component needs to be judged in relation to the review finding. For instance, where a finding represents 'mid-level' theory regarding a phenomenon, it may be important that this is backed by considerable data and that the fit between the data from the primary studies and the review finding is clear. Concerns regarding adequacy of data and coherence may therefore be particularly critical in making an overall CERQual assessment for this finding
  - The data contributing to a review finding may come from studies that are assessed to have different levels of concern in relation to a CERQual component. This variation can be captured in three ways: (1) make a judgment that captures the highest level of concern for the component; (2) make a judgment that captures the lowest level of concern for the component; or (3) make a judgment that captures the "middle ground" for the component.
- For example, a synthesis of parents' views and experiences of communication for childhood vaccination includes a finding that parents felt that the information that they received about vaccination was unbalanced or one sided. This finding was based on three ethnographic studies, two assessed as having no or very minor concerns and one assessed as having moderate concerns regarding methodological limitations; and two focus group studies, both assessed as having moderate concerns regarding methodological limitations. The synthesis authors decide to give an umbrella assessment of 'moderate concerns' for the methodological limitations component
  - The initial assessment of coherence may prompt changes in the way that a review finding is conceptualised and described if, for example, it becomes clear that the finding would make more sense as two separate findings. Where this occurs, the assessments for all of the CERQual components may then need to be revisited for the finding/s
- Recommended standard phrases for describing the assessment for each CERQual component and the overall assessment are provided in Additional file 4
- You should strive to be consistent across review findings in a synthesis in assessing the extent of concerns regarding each CERQual component. Consistency across syntheses is harder to achieve and it is more important to be explicit and transparent regarding judgements so that the rationale for these are clear to users

changes to this assessment to address concerns about relevance or other CERQual components.

Currently, there is no established order in which to assess CERQual components. Thus far, review teams have typically begun with methodological limitations, but experience suggests that an assessment is an iterative process involving moving between the assessment for each component and the overall judgement of confidence in the evidence. Further research and experience is needed to ascertain if there is an optimal order of applying the components.

The overall CERQual assessment for each review finding should be explained in a transparent manner, preferably in a Summary of Qualitative Findings table that includes a narrative explanation of the CERQual assessment (see below). Where a review finding hypothesises a connection or pathway between factors, the accompanying assessment of confidence should take into account the strength of the evidence for this proposed pathway.

### Creating a CERQual Evidence Profile and a Summary of Qualitative Findings (SoQF) table

The CERQual Evidence Profile and SoQF table both intend to provide a structured summary of the review findings and of the information contributing to the CERQual assessment for each finding, with the Evidence Profile providing more detail than the SoQF table.

Developing these outputs encourages the review team to carefully consider what constitutes a finding in the context of the synthesis and how to express each review finding clearly. It also helps to ensure that judgements underlying the CERQual assessments are transparent.

We suggest that review findings and CERQual assessments be presented in three 'layers' in a synthesis: firstly, as *full review findings* reported in the 'Findings' or 'Results' section of the synthesis. This is the most detailed presentation of each review finding and may include verbatim data extracts from the studies contributing to the review finding as well as the CERQual assessments for each review finding. Secondly, in the *Evidence Profile*, which includes summaries of the review findings, information on the judgments for each CERQual component underlying the overall CERQual assessment as well as the overall assessment and its explanation. The Evidence Profile is where members of the review team can make explicit and transparent their judgments underlying each CERQual component assessment as well as the judgments underlying the overall assessment. Some users of review findings may find this level of detail helpful. Thirdly, in the *SoQF table*, which is a shorter version of the Evidence Profile and includes summaries of the review findings, the overall CERQual assessments, an explanation of each CERQual assessment and references to the studies contributing to each review finding. The

SoQF table is most useful to users of the review findings (for example, guidelines panels).

In the sections below, we describe in more detail the content of the Evidence Profile and SoQF table.

**CERQual Evidence Profile** A CERQual Evidence Profile is used to provide information on all the component assessments that contribute to the overall CERQual assessment for each review finding. The Evidence Profile includes five elements (Tables 3 and 4): (1) a summary of each review finding; (2) an explanation of the assessment made for each CERQual component for each review finding. We have developed standard phrases for reporting the results of these assessments (see above and Additional file 4); (3) an overall CERQual assessment for each individual review finding; (4) an explanation of the overall CERQual assessment; (5) reference to the studies contributing data to the review finding, including clarification of the contexts in which those studies were conducted.

**Summary of Qualitative Findings tables** A Summary of Qualitative Findings table is used to summarise the key findings of the synthesis and to facilitate the understanding and use of review findings—it may often be the entry point to the full review but is not a fully 'standalone' product as it cannot capture all of the nuances of a full review finding. A SoQF table is the final output of the process of making a CERQual assessment and includes three elements: (1) a summary of each review finding; (2) an overall CERQual assessment for each review finding; (3) reference to the studies contributing data to the review finding, including clarification of the contexts in which those studies were conducted. A SoQF table may also include an explanation of the overall CERQual assessment. As far as possible, a SoQF table should include only findings emerging from the analysis and should not include related background information, for instance on the methods used in the synthesis. Tables 6 and 7 present examples of Summary of Qualitative Findings tables.

Review teams need to decide how best to organise review findings, together with their CERQual assessments, in the SoQF table. In doing so, they should be guided by what would make most sense to those likely to use these findings. Options include organising review findings by broad themes or by settings or population groups; from higher to lower priority in relation to the review question; and by level of confidence, starting with the highest confidence review findings.

CERQual is intended to be applied to all review findings reported in a synthesis. It is preferable for all review findings to be reported in the SoQF table regardless of their level of confidence. However, as determined by the requirements of particular audiences, review teams may decide to include only the highest priority review findings in a SoQF table in the main text of the review and to include the remaining findings in a supplementary SoQF table. It may also be useful to include the Evidence Profile as a supplementary table. The following types of review findings should generally be included in the main SoQF table: review findings that relate most closely to the review question; review findings relevant to the decision making process for which the synthesis was

**Table 6** CERQual Summary of Qualitative Findings table—Example A

Objective: To synthesise qualitative and quantitative evidence on the mistreatment of women during childbirth in health facilities.
Perspective: Experiences and attitudes of stakeholders in any country about the mistreatment of women during childbirth

| Summary of review finding | Studies contributing to the review finding | CERQual assessment of confidence in the evidence | Explanation of CERQual assessment |
|---|---|---|---|
| 1. Use of force: Women across the world reported experiencing physical force by health providers during childbirth. In some cases, women reported specific acts of violence committed against them during childbirth, but women often referred to these experiences in a general sense and alluded to beatings, aggression, physical abuse, a rough touch and use of extreme force. Pinching, hitting and slapping, either with an open hand or an instrument were the most commonly reported specific acts of physical violence. | 6, 9, 10, 13, 21, 61, 67, 68, 73, 75, 77, 80, 84, 86, 87, 91, 96, 97 | High confidence | 15 studies with moderate methodological limitations. Data from 10 countries across all geographical regions, but predominantly sub-Saharan Africa. No or very minor concerns about coherence and adequacy. |
| 2. Physical restraint: Women reported physical restraint during childbirth through the use of bed restraints and mouth gags. | 86, 97 | Very low confidence | Two studies (Tanzania and Brazil) with moderate methodological limitations. Limited, thin data from 2 countries. Minor concerns about coherence but limited data available. |

Review findings taken from [12] and adapted to fit the context of this article. The review findings presented here are drawn from the wider thematic synthesis undertaken for this review. The themes identified were summarised into summaries of review findings, as illustrated in this table. The methods are described in more detail in [12]

**Table 7** CERQual Summary of Qualitative Findings table—Example B

Objective: To identify, appraise and synthesise qualitative research evidence on the barriers and facilitators to the implementation of lay health worker programmes for maternal and child health

Perspective: Experiences and attitudes of stakeholders about lay health worker programmes in any country

Included programmes: Programmes that were delivered in a primary or community healthcare setting; that intend to improve maternal or child health; and that had used any type of lay health worker, including community health workers, village health workers, birth attendants, peer counsellors, nutrition workers and home visitors

| Summary of review finding | Studies contributing to the review finding | CERQual assessment of confidence in the evidence | Explanation of CERQual assessment |
|---|---|---|---|
| 1. While regular salaries were not part of many programmes, other monetary and non-monetary incentives, including payment to cover out-of-pocket expenses and 'work tools' such as bicycles, uniforms or identity badges, were greatly appreciated by lay health workers. | 2, 5, 11, 12, 22, 29 | Moderate | Minor concerns regarding methodological limitations, relevance, coherence and adequacy. |
| 2. Some unsalaried lay health workers expressed a strong wish for regular payment. | 5, 13 | Low | Moderate concerns regarding relevance and serious concerns regarding adequacy of data. |
| 3. Lay health workers, particularly those working in urban settings, reported difficulties maintaining personal safety when working in dangerous settings or at night. | 3, 15, 16, 25, 31 | Moderate | Moderate methodological limitations and moderate concerns regarding adequacy of data. |

Review findings taken from [14] and adapted to fit the context of this article. The review findings presented here are drawn from the wider thematic synthesis undertaken for this review. The themes identified were summarised into evidence statements, as illustrated in this table. The methods are described in more detail in [14]

commissioned; review findings that are more novel or are unexpected given the current level of knowledge; and review findings that are more likely to affect practice or policy.

Review findings that are assessed as very low confidence should also be included in a SoQF table, and review finding 2 in Tables 3 and 6 provides an example of this. CERQual may help to distinguish between situations in which our confidence in a review finding is lowered because there is little data available, and situations in which our confidence is lowered for other reasons. The former relates to adequacy of data while the latter relates to the other CERQual components. Further, where data underlying a review finding are not sufficiently rich or only come from a small number of studies or participants, this should not be taken to mean that this phenomenon is uncommon or unlikely. Rather, it means that we do not have adequate data on the phenomenon of interest to feel confident about the finding, and this may be because little research has been conducted on this issue. In cases where a qualitative evidence synthesis finds no data regarding a (pre-specified) phenomenon, or an aspect of a phenomenon, it may be important to report in both the findings and the SoQF table that no data were found and therefore that no CERQual assessment could be made.

In both SoQF tables and Evidence Profiles, the explanation of each overall CERQual assessment is important to the end user as this shows how the final assessment was reached and increases the transparency of the judgement process. Further, where end users seek evidence for a question that differs slightly from the original synthesis question (for instance, addressing a different practice setting), they are able to see clearly how the

assessment has been made and adjust their own confidence in the review finding accordingly.

**Implications of the CERQual component and overall assessments for future primary research** In other papers in the series, we discuss the implications for future primary research associated with each CERQual component assessment [5–8]. Where an overall CERQual assessment indicates that our confidence in a review finding is moderate, low or very low, this also indicates that more and/or better-conducted primary research is needed. Suggestions for new research should also aim to address the specific concerns identified in relation to each CERQual component.

## Conclusions

This paper has discussed the process for making an overall CERQual assessment of confidence in a review finding and outlined how to create an Evidence Profile and a Summary of Qualitative Findings table. The Evidence Profile and SoQF Table can, in turn, feed into decision making processes on health, social or other interventions.

To inform a decision on whether to recommend or implement an intervention or policy, findings from a qualitative evidence synthesis regarding, for example, the acceptability and feasibility of an intervention, typically need to be presented alongside findings on other factors, such as the benefits and harms of an intervention and its resource use. Evidence-to-decision frameworks set out in a structured and transparent way what is known about each factor (benefits and harms, acceptability, feasibility, etc.), and can be used to record the judgments of those making the decision regarding each factor as

well as how each factor contributed to the decision [24]. SoQF tables are a good source of information for populating evidence-to-decision tables as the SoQF provides short summaries of each review finding as well as assessments of confidence in each finding. The GRADE-CERQual project group will provide further guidance in the future on using CERQual outputs in decision making processes.

As with the GRADE approach for assessing the certainty of evidence of effectiveness [22], GRADE-CERQual should not be seen as a mechanistic approach for assessing how much confidence to place in findings from qualitative evidence syntheses. An assessment of confidence is a subjective process, and CERQual does not remove the need for judgement or intend to suggest that an objective assessment of confidence can be made.

### Open peer review
Peer review reports for this article are available in Additional file 5.

### Additional files

**Additional file 1:** Key definitions relevant to CERQual. (PDF 617 kb)

**Additional file 2:** Deciding whether to apply CERQual to a review finding—interpretive or explanatory level findings. (PDF 637 kb)

**Additional file 3:** Guidance on applying CERQual to a qualitative evidence synthesis conducted by another review team. (PDF 650 kb)

**Additional file 4:** Recommended standard phrases for describing the assessment for each CERQual component and the overall assessment. (PDF 598 kb)

**Additional file 5:** Open Peer Review reports. (PDF 154 kb)

### Acknowledgements
Our thanks for their feedback to those who participated in the GRADE-CERQual Project Group meetings in January 2014 or June 2015 or gave comments to the paper: Elie Akl, Heather Ames, Zhenggang Bai, Rigmor Berg, Kam-wa Chan, Jackie Chandler, Karen Daniels, Hans de Beer, Kenny Finlayson, Bela Ganatra, Susan Munabi-Babigumira, Andy Oxman, Tomas Pantoja, Hector Pardo-Hernandez, Vicky Pileggi, Kent Ranson, Rebecca Rees, Holger Schünemann, Anna Selva, Elham Shakibazadeh, Birte Snilstveit, James Thomas, Hilary Thomson, Judith Thornton, Josh Vogel and Kieran Walsh. Thanks also to Sarah Rosenbaum for developing the figures used in this series of papers and to members of the GRADE Working Group for their input. The guidance in this paper has been developed in collaboration and agreement with the GRADE Working Group (http://www.gradeworkinggroup.org).

### Funding
This work, including the publication charge for this article, was supported by funding from the Alliance for Health Policy and Systems Research, WHO (http://www.who.int/alliance-hpsr/en). Additional funding was provided by the Department of Reproductive Health and Research, WHO (www.who.int/reproductivehealth/ about_us/en/); Norad (Norwegian Agency for Development Cooperation: www.norad.no), the Research Council of Norway (www.forskningsradet.no); and the Cochrane Methods Innovation Fund. SL is supported by funding from the South African Medical Research Council (www.mrc.ac.za). The funders had no role in study design, data collection and analysis, preparation of the manuscript or the decision to publish.

### Availability of data and materials
Additional materials are available on the GRADE-CERQual website (www.cerqual.org)

To join the CERQual Project Group and our mailing list, please visit our website: http://www.cerqual.org/contact/. Developments in CERQual are also made available via our Twitter feed: @CERQualNet.

### About this supplement
This article has been published as part of *Implementation Science* Volume 13 Supplement 1, 2018: Applying GRADE-CERQual to Qualitative Evidence Synthesis Findings. The full contents of the supplement are available online at https://implementationscience.biomedcentral.com/articles/supplements/volume-13-supplement-1.

### Authors' contributions
All authors participated in the conceptual design of the CERQual approach. SL, MB and AR wrote the first draft of the manuscript. All authors contributed to the writing of the manuscript. All authors have read and approved the manuscript.

### Ethics approval and consent to participate
Not applicable. This study did not undertake any formal data collection involving any humans or animals.

### Consent for publication
Not applicable.

### Competing interests
The authors declare that they have no competing interests.

### Author details
[1]Norwegian Institute of Public Health, Oslo, Norway. [2]Health Systems Research Unit, South African Medical Research Council, Cape Town, South Africa. [3]UNDP/UNFPA/UNICEF/WHO/World Bank Special Programme of Research, Development and Research Training in Human Reproduction, Department of Reproductive Health and Research, World Health Organization, Geneva, Switzerland. [4]Department of Health Management and Economics, School of Public Health, Tehran University of Medical Sciences, Tehran, Iran. [5]Information, Evidence and Research Department, Eastern Mediterranean Regional Office, World Health Organization, Cairo, Egypt. [6]Division of Social and Behavioural Sciences, School of Public Health and Family Medicine, University of Cape Town, Cape Town, South Africa. [7]European Centre for Environment and Human Health, University of Exeter Medical School, Exeter, UK. [8]School of Social Sciences, Bangor University, Bangor, UK. [9]School of Health & Related Research (ScHARR), University of Sheffield, Sheffield, UK. [10]University of North Carolina at Chapel Hill, Chapel Hill, NC, USA. [11]Uni Research Rokkan Centre, Bergen, Norway.

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
