## [Open Peer Review reports. (PDF 154 kb) · Implementation Science : IS]

Open Peer Review reports

Original Submission		
21 Oct 2016	Submitted	Original Manuscript
20 Jan 2017	Reviewed	Reviewer Report - Monika Kastner
		Summary This study is one of a series of papers describing the development of guidance for GRADE-CERQual (Confidence in Evidence from qualitative evidence synthesis); CERQual is being developed by the GRADE working group, and includes 4 components for assessing how much confidence can be placed in findings from qualitative reviews. This particular paper involves the overall assessment of confidence in a review finding and creating a CERQual Evidence Profile and Summary of Qualitative Findings table. Overall, this is an interesting and important paper. The paper is generally well done, but there are some aspects of the methods that could be clarified. I have provided some suggestions for clarity that authors can consider to strengthen the manuscript. Background * On page 5, lines 28-29, could authors elaborate on what they mean by: "How review findings are defined and presented...?"* The 4th factor assessing confidence in the review findings is a bit unclear: what do authors mean by the "context specified in the review question"?* The sentence starting on line 46 (page 5) re: "Making an overall assessment of confidence in a review finding..." is a bit unclear* It's not clear what authors mean by "subsequent papers in the series describe how to assess each CERQual component" - what component?* Also, it might be helpful to have a bit of an intro of all the papers in the series, and then number each papers (rather than stating that "subsequent" papers) and to indicate the number is the current manuscript; this might help place all the papers in context* The objective (page 5: lines 51-55), and the aim (page 6: lines 35-50) is separated by two paragraphs that don't really seem to fit - there should be one objectives statement* On Page 6 (lines 8-13), authors mention that this paper is the endpoint of CERQual assessment - this should be mentioned earlier (and tie in with previous and subsequent papers so the reader knows how to place this paper in the context of the others)* The paragraph prior to the "aim" should be moved to the Methods* In the "aims" paragraph, rather than calling these "topics", would a better term be processes?* Overall, it would be helpful to describe why CERQual is needed and how this will overcome the problems - for example, what are the specific methodological issues that warrant the use of such guidance, and this should be referenced with examples Methods * Although authors mention that the methods used to develop CERQual is described in the first paper, there should be just a bit more detail here... for example:o What literature was searched and what were the date parameters? What definitions?

- o How many qualitative experts were engaged?
- o Who were involved in the consensus group meetings? How was consensus reached and on what specifically?
- o How was the qualitative synthesis community engaged?
- o With what qualitative evidence syntheses were the CERQual tested and how?

Results and implications

- * It would be helpful to provide some instructions on how to use this paper - for example, the sub-headings - are these process steps?
- * Is there evidence for the benefits of summaries of review findings? (page 7, line 41)
- * How was it determined what are "poorly" or "more appropriately" written summaries?
- * What are the "types and levels" of review findings for which CERQual can be applied?
- * In the "Making an overall CERQual assessment of confidence in each individual review finding" section, it would be helpful to provide an example for each of the 4 factors to be used for the assessment
- * Overall, it might be helpful to provide a visual representation of the process (and who should be involved at every stage) of the CERQual process to help place the overall guidance into perspective for those who will use this
- * Are there any plans to validate the CERQual?
- * Figure 1 is not clear

17 Mar 2017

Reviewed

Reviewer Report - Helen Smith

General comments

Thank you for the opportunity to review this paper. I believe the series is valuable and timely for authors and potential users of qualitative evidence syntheses. As with my previous review, my comments mainly relate to making the concept and procedures accessible to a wide readership. An overall concern is the current format of the paper. It is currently written as a methodological paper in a standard research article format, yet it reads as a descriptive 'how to' guide. The methods as they are currently described do not seem to describe any 'research methods', and the 'results' don't represent findings (rather they describe processes). I suppose there are two options here: a) to keep the format of a research article and make more of the user testing as the method used and integrate more findings from the user testing in the paper, or b) to format the paper as a 'how to do CERQual' type paper, which describes the detailed processes involved (which the current paper seems to do but within a research article format). In my comments below I have tried to highlight where findings from the user testing could be integrated, which in my view would strengthen the paper.

Specific comments

1. Page 5 lines 22-28: the term 'analytic output' is only likely to be understood by a minority of readers, and the explanation which refers to 'an aspect of a phenomenon' is a bit abstract and could be better described. The most common analytic outputs of a QES are usually themes or categories, or a theory or contribution to a theory. These are terms that readers would readily grasp.
2. Page 5 lines 51-55 and page 6 line 35-39: Here the authors state the purpose of the paper is to describe and discuss the process of making an overall assessment and how to create a CERQual evidence profile and summary of findings table. If this is the focus, then I don't think the paper should be written as a standard research article; there are no 'findings' and no research methods have been employed (see general comment above).
3. Page 7 lines 3-18: As per my general comment, I don't think what is described here constitutes 'methods' or any recognisable research process. What definitions were searched for any why and how were these used in the process of developing the CERQual approach? Which experts were consulted

and on what aspects of the method or guidance? Then user testing is mentioned, but not as a central methodology. If the paper remains as a research article then I would encourage the authors to make more of the user testing, describe in more detail how it was carried out (what methods were used), by whom (reviewers and users or just reviewers), on which reviews (and how were they selected) etc. Then in the results, the authors could explicitly draw on examples from the user testing.

4. Page 7 line 28: this part of the results section describes processes. Can the authors bring in real examples from the user testing? For example, based on the user testing, was moving from review findings to a summary of a finding straightforward?, how was it done?, what were the experiences of those doing the testing? what challenges did the users in the test face? Table 1 is very dense, hard to navigate so much text and description. What is it the reader should be readily grasping from this? It would be a great resource for the CERQual website, to provide guidance to reviewers carrying out a CERQual assessment, but in a paper I question the value of such detail. If the paper was written up with actual findings from the user testing, the authors could provide real examples of SoF tables and critique what works well, less well, what to look out for (in place of tables 1 and 2)

5. Page 8 lines 15-18: this paragraph suggests review teams should decide whether it is appropriate to apply CERQual to all findings- again it would be useful to give examples from the user testing of when it is and isn't appropriate to apply it.

6. Later on page 8 the authors suggest the review team needs to decide at what level of finding to apply CERQual - again I feel some examples from the reviews included in the user testing would be helpful.

7. Page 9 lines 55-58: the distinction between important concerns and very minor concerns could also be illustrated (and therefore made clearer) using examples from reviews used in the user testing.

8. Page 10-11: CERQual assessments are described as judgements and the paper describes who should do the assessments. Again, I think the paper could be strengthened with examples from the user testing - how often did disagreements emerge, on what aspects, and examples of how this was dealt with.

9. Page 12: the authors present suggestions for further research on this page. In a standard research article implications for research would usually be described in the discussion section, which again makes me think the paper needs a bit of restructuring if it is to fit this format.

10. Page 12, lines 52-54: the authors state 'experience suggests' assessment is iterative - why not present findings from the user test here? How were the components dealt with in reviews included in the user testing?

11. Page 14: where the evidence profiles are described it would help if the examples in tables 3 and 4 could be critiqued and the authors could offer specific guidance (drawing on the user test findings) for each component.

12. Table 8 comprises three paragraphs of 'guidance' and it's not clear why it is presented in a table or how it is a 'finding'. Could be a useful resource on the website, for those carrying out a CERQual assessment.

13. Page 16: The conclusion again reiterates that the paper is about the process for making a CERQual assessment, yet it is written up in the format of a research article. I would encourage the authors to consider what they want to say in this paper - whether they want to provide a guidance document describing 'how to do' an assessment, or whether they want to provide the results of user testing the approach on a series of reviews.

26 Jun 2017

Author responded

Author comments - Simon Lewin

Peer reviewer comments	Responses
Reviewer #1:	
Summary: This study is one of a series of papers describing the development of guidance for GRADE-CERQual (Confidence in Evidence from qualitative evidence synthesis); CERQual is being	No response needed

developed by the GRADE working group, and includes 4 components for assessing how much confidence can be placed in findings from qualitative reviews. This particular paper involves the overall assessment of confidence in a review finding and creating a CERQual Evidence Profile and Summary of Qualitative Findings table. Overall, this is an interesting and important paper. The paper is generally well done, but there are some aspects of the methods that could be clarified. I have provided some suggestions for clarity that authors can consider to strengthen the manuscript.	
Background: * On page 5, lines 28-29, could authors elaborate on what they mean by: "How review findings are defined and presented...?"	We have added additional text to this paragraph to clarify what we mean by 'a review finding' and what we mean by 'the phenomenon of interest'. We have also elaborated on different approaches to qualitative evidence synthesis, in relation to their outputs, noting that "...the aims of different approaches to QES range from identifying and describing key themes, to seeking more generalizable or interpretive explanations that can be used for building theory."
* The 4th factor assessing confidence in the review findings is a bit unclear: what do authors mean by the "context specified in the review question"?	Please see additional text to clarify what we mean by 'context' (Page 5): "...the relevance of the data from the primary studies supporting a review finding to the context (perspective or population, phenomenon of interest, setting) specified in the review question."
* The sentence starting on line 46 (page 5) re: "Making an overall assessment of confidence in a review finding..." is a bit unclear	Please see the revised version which clarifies this sentence: "Making an overall assessment of confidence in a review finding involves moving from the judgements made for each CERQual component to a final assessment. The overall assessment of confidence in a review finding that takes into account the concerns identified in relation to each of the four components."
* It's not clear what authors mean by "subsequent papers in the series describe how to assess each CERQual component" - what component?	We have revised the text to clarify this., as follows: "Subsequent papers in the series describe how to assess each CERQual component (methodological limitations, coherence, adequacy, and relevance)."
* Also, it might be helpful to have a bit of an intro of all the papers in the series, and then number each papers (rather than stating that "subsequent" papers) and to indicate the number is the current manuscript; this might help place all the papers in context	Thanks for this suggestion which we have implemented across the series.

* The objective (page 5: lines 51-55), and the aim (page 6: lines 35-50) is separated by two paragraphs that don't really seem to fit - there should be one objectives statement	We agree that there should be one statement of the aims of the paper. We have therefore deleted the first mention of the objective of the paper and created a new section in the paper ('Aim') that describes the aim of this paper as well as how it relates to the other papers in this series. The section titled 'Aim' is standard across papers 2 to 7 in this series.
* On Page 6 (lines 8-13), authors mention that this paper is the endpoint of CERQual assessment - this should be mentioned earlier (and tie in with previous and subsequent papers so the reader knows how to place this paper in the context of the others)	Please see the revisions described above that reorganize the Background section to address this comment.
* The paragraph prior to the "aim" should be moved to the Methods	This paragraph ("A key part of communicating...") has been moved to the newly created 'Aim' section, where it helps to explain the importance of the Summary of Qualitative Findings table. Developing and populating a Summary of Qualitative Findings Table is one of the aims of the paper, and we believe that this paragraph helps to explain this.
* In the "aims" paragraph, rather than calling these "topics", would a better term be processes?	We have revised "topics" to "processes".
* Overall, it would be helpful to describe why CERQual is needed and how this will overcome the problems - for example, what are the specific methodological issues that warrant the use of such guidance, and this should be referenced with examples	We believe that this is addressed adequately in Paper 1 of this series. In this paper, we state the following in paragraph 1 of the Background section: "The importance of assessing confidence in qualitative evidence is discussed in the first paper in this series [1]." No revisions made.

Methods: * Although authors mention that the methods used to develop CERQual is described in the first paper, there should be just a bit more detail here... for example:  o What literature was searched and what were the date parameters? What definitions? o How many qualitative experts were engaged? o Who were involved in the consensus group meetings? How was consensus reached and on what specifically? o How was the qualitative synthesis community engaged? o With what qualitative evidence syntheses were the CERQual tested and how? 	It was agreed with the Implementation Science editor for this series that a detailed description of the methods used to develop CERQual would be included in Paper 1, and that an abbreviated methods section would be included in the other papers, to reduce repetition and save space. We have now provided further details in this paper of the methods used for this aspect of the CERQual approach, but readers will need to navigate to Paper 1 for a full methods description.
Results and implications: * It would be helpful to provide some instructions on how to use this paper - for example, the sub-headings - are these process steps?	Thanks for this suggestion which we have addressed in two ways: firstly, at the end of the Background section, we have indicated to the reader that the subheadings reflect the processes involved in making an overall assessment of confidence. Secondly, we have added a 'user guide' table to paper 1 (see additional figure) which provides instructions on how each of the papers in this series can be used.
* Is there evidence for the benefits of summaries of review findings? (page 7, line 41)	We have listed the benefits to include (1) identifying central ideas of each finding; (2) reflexive discussions about each finding; and (3) a starting point for creating an Evidence Profile and Summary of Qualitative Findings (SoQF). We have now also added a fourth benefit: "Fourth, the summaries of review findings in the CERQual Evidence Profiles and SoQF are used in the guideline process as tools to clearly and concisely present the evidence, and confidence in this evidence, to the guideline panel." This list of benefits are based on our collective experience of creating and using these summary review findings across a range of reviews and decision making processes.
* How was it determined what are "poorly" or "more appropriately" written summaries?	In Table 2, we were trying to show how summaries of review findings could be improved to make the finding as clear and explicit as

	possible. We have now revised both the reference to the Table in the text of the paper and the Table title to reflect this. The Table is now titled 'Examples of summary review findings, including how these might be improved' and we have added a footnote explaining that an assessment of whether a finding is sufficiently clear and explicit involves a judgment.
* What are the "types and levels" of review findings for which CERQual can be applied?	We have clarified that the two sentences following this statement are explaining what we mean by "types and levels" (page 8).
* In the "Making an overall CERQual assessment of confidence in each individual review finding" section, it would be helpful to provide an example for each of the 4 factors to be used for the assessment	We believe that the description of the four levels of concern used for the assessment are sufficiently well described in the bullet points. In addition, the individual component papers in this series explain how to make the assessment for that component. However, we have added a sentence to clarify that the assessment should be described using one of the four levels of concern.
* Overall, it might be helpful to provide a visual representation of the process (and who should be involved at every stage) of the CERQual process to help place the overall guidance into perspective for those who will use this	Thanks for this helpful suggestion. As noted above, we have added an additional figure that describes how each paper in the series can be used, and by whom, to paper 1.
* Are there any plans to validate the CERQual?	We do not currently have plans to validate CERQual in the way that one might validate a psychometric instrument as this would not be appropriate or useful given the nature of the approach used within CERQual. However, we aim to improve CERQual continuously through feedback from users and discussions within the CERQual Project Group.
* Figure 1 is not clear	Based on feedback from several reviewers, we have revised Figure 1 to make it clearer and more useful.
Reviewer #2:	
General comments: Thank you for the opportunity to review this paper. I believe the series is valuable and timely for authors and potential users of qualitative evidence syntheses. As with my previous review, my comments mainly relate to making the concept and procedures accessible to a wide readership. An overall concern is the current format of the paper. It is currently written as a methodological paper in a standard research article format, yet it reads as a descriptive 'how to' guide. The methods as they are currently described do not seem to describe any 'research	We have made a number of changes to address these comments: 1. In consultation with the journal, we have changed the format for all of the papers, including the main headings, to better reflect that these are 'how to' papers rather than papers reporting empirical research findings 2. We have changed titles of the papers as follows 'Applying GRADE-CERQual to qualitative evidence synthesis findings – paper X: [title of paper]'. We believe this will indicate more clearly

methods', and the 'results' don't represent findings (rather they describe processes). I suppose there are two options here: a) to keep the format of a research article and make more of the user testing as the method used and integrate more findings from the user testing in the paper, or b) to format the paper as a 'how to do CERQual' type paper, which describes the detailed processes involved (which the current paper seems to do but within a research article format). In my comments below I have tried to highlight where findings from the user testing could be integrated, which in my view would strengthen the paper.	to readers that these are 'how to' papers and will also make it easier to navigate through the series
Specific comments: 1. Page 5 lines 22-28: the term 'analytic output' is only likely to be understood by a minority of readers, and the explanation which refers to 'an aspect of a phenomenon' is a bit abstract and could be better described. The most common analytic outputs of a QES are usually themes or categories, or a theory or contribution to a theory. These are terms that readers would readily grasp.	We revised the text to clarify what we mean by "analytic output". We explain what we mean by phenomenon in the following sentence, and have revised it for better clarity.
2. Page 5 lines 51-55 and page 6 line 35-39: Here the authors state the purpose of the paper is to describe and discuss the process of making an overall assessment and how to create a CERQual evidence profile and summary of findings table. If this is the focus, then I don't think the paper should be written as a standard research article; there are no 'findings' and no research methods have been employed (see general comment above).	As noted above, we have changed the format for all of the papers, including the main headings, to better reflect that these are 'how to' papers rather than papers reporting empirical research findings
3. Page 7 lines 3-18: As per my general comment, I don't think what is described here constitutes 'methods' or any recognisable research process. What definitions were searched for any why and how were these used in the process of developing the CERQual approach? Which experts were consulted and on what aspects of the method or guidance? Then user testing is mentioned, but not as a central methodology. If the paper remains as a research article then I would encourage the authors to make more of the user testing, describe in more detail how it was carried out (what methods were used), by whom (reviewers and users or just reviewers), on which reviews (and how were they selected) etc. Then in the results, the authors could explicitly draw on examples from the user testing.	Please see the responses above. In addition, a more detailed description of 'How CERQual was developed' is included in paper 1 in the series.

4. Page 7 line 28: this part of the results section describes processes. Can the authors bring in real examples from the user testing? For example, based on the user testing, was moving from review findings to a summary of a finding straightforward?, how was it done?, what were the experiences of those doing the testing? what challenges did the users in the test face? Table 1 is very dense, hard to navigate so much text and description. What is it the reader should be readily grasping from this? It would be a great resource for the CERQual website, to provide guidance to reviewers carrying out a CERQual assessment, but in a paper I question the value of such detail. If the paper was written up with actual findings from the user testing, the authors could provide real examples of SoF tables and critique what works well, less well, what to look out for (in place of tables 1 and 2)	Please see responses above. This paper is not intended to report the details of user-testing. However, the 'how to' guidance that we provide and the examples that we include are mostly drawn from 'actual' qualitative evidence syntheses in which CERQual has been applied or from issues that arose during CERQual training workshops. We agree that Table 1 was difficult to use and have now greatly changed this. Much of the text has been integrated into the main text of the paper, allowing us to shorten and simplify the content of the table. We have also re-ordered the remaining text, removed duplication and improve the structure so that the table provides clearer guidance. Examples of summary review findings are provided in Table 2, and these are drawn (with adaptation) from three published syntheses that have applied CERQual.
5. Page 8 lines 15-18: this paragraph suggests review teams should decide whether it is appropriate to apply CERQual to all findings-again it would be useful to give examples from the user testing of when it is and isn't appropriate to apply it.	We have attempted to clarify this on page 8 as follows: "In general, the review team would assess all review findings emerging from a QES, but there may be circumstances in which this is not feasible or appropriate. For instance, some of the findings from a QES may be particularly relevant to a decision making process and the review team may therefore choose to apply CERQual to those findings only." We then go on to explain why it may be important to apply CERQual to as many of the review findings as possible. Unfortunately we do not have a worked example at this stage of the application of CERQual to a specific group of findings (rather than all findings) from a QES. We also want to mention that this paper, and the others in the series, is not intended to be a user-testing paper. This papers aims to "discuss the process for making an overall assessment of confidence in a review finding and to outline how to create a CERQual Evidence Profile and a Summary of Qualitative Findings table". In addressing this aim, we draw on both 'real' and hypothetical examples but we have not formally

	user tested the CERQual approach or assessed the extent to which users of CERQual agree on their judgements regarding concerns or confidence in the evidence. We therefore did not intend to imply that this paper would report such testing. We have checked the language throughout to make sure that this is clear.
6. Later on page 8 the authors suggest the review team needs to decide at what level of finding to apply CERQual - again I feel some examples from the reviews included in the user testing would be helpful.	We agree that this is also an important issue, but unfortunately we don't yet have examples of the application of CERQual to different 'levels' of findings within a review. Additional file 1 gives more guidance on this, and includes an example finding from a review that we have adapted to explain what might be done with different levels of findings. Work is currently underway to apply CERQual to findings from a meta-ethnography and we therefore anticipate having 'real' worked examples in the near future.
7. Page 9 lines 55-58: the distinction between important concerns and very minor concerns could also be illustrated (and therefore made clearer) using examples from reviews used in the user testing.	Several reviewers have made this point across the papers in this series. We provide several examples of what constitutes important or minor concerns in Tables 3 and 4 (CERQual evidence profiles from published reviews) – we have now made this clearer in the text. In addition, we have added examples to each of the papers describing a CERQual component (papers 3, 4, 5 and 6 in the series) and we believe that this will help users understand the distinction between different levels of concerns.
8. Page 10-11: CERQual assessments are described as judgements and the paper describes who should do the assessments. Again, I think the paper could be strengthened with examples from the user testing - how often did disagreements emerge, on what aspects, and examples of how this was dealt with.	Please see our response above regarding 'user testing'.
9. Page 12: the authors present suggestions for further research on this page. In a standard research article implications for research would usually be described in the discussion section, which again makes me think the paper needs a bit of restructuring if it is to fit this format.	Thanks for this suggestion and please see our response above regarding the structure of the paper, which has now been changed. Paper 1 in this series describes the broad research agenda for CERQual (Paper 1, Table 4). In addition, some specific research implications are included in the component and overall assessment papers. We have chosen to describe these specific research questions in the section of the paper in which they arise rather than gathering them into a single section or table as we judged that these questions would be easier to understand in context.

10. Page 12, lines 52-54: the authors state 'experience suggests' assessment is iterative - why not present findings from the user test here? How were the components dealt with in reviews included in the user testing?	Please see our response above regarding user testing. We feel that the current text captures adequately our state of knowledge regarding the assessment of CERQual components: "Currently there is no established order in which to assess CERQual components. Thus far, review teams have typically begun with methodological limitations, but experience suggests that an assessment is an iterative process involving moving between the assessment for each component and the overall judgement of confidence in the evidence. Further research and experience is needed to ascertain if there is an optimal order of applying the components." Because the process is iterative, the starting point, in terms of component, or order of considering the components, is not really of consequence. The main issue here is for users to understand that they need to look holistically at all of the concerns for all of the components in order to decide whether to 'grade down' the confidence in the finding. We have added an additional figure (Figure 2) that we hope illustrates that all CERQual components contribute to an overall assessment of confidence.
11. Page 14: where the evidence profiles are described it would help if the examples in tables 3 and 4 could be critiqued and the authors could offer specific guidance (drawing on the user test findings) for each component.	Specific guidance for assessing each of the CERQual components is included in each of the individual component papers (papers 3, 4, 5 and 6 in the series). We do not think that it is appropriate to critique our own examples in Tables 3 and 4, which have anyway been adapted for the context of this series. No revisions made.
12. Table 8 comprises three paragraphs of 'guidance' and it's not clear why it is presented in a table or how it is a 'finding'. Could be a useful resource on the website, for those carrying out a CERQual assessment.	We have reviewed the content of Table 8 and feel that it makes an important point about situations where data are not available on an aspect of the phenomenon of interest. However, we agree that this content could be written more succinctly and we have done this and integrated it into the main text of the paper. We then deleted Table 8.
13. Page 16: The conclusion again reiterates that the paper is about the process for making a CERQual assessment, yet it is written up in the format of a research article. I would encourage the authors to consider what they want to say in	Please see our responses above regarding the structure of the paper and how it has been framed.

this paper - whether they want to provide a guidance document describing 'how to do' an assessment, or whether they want to provide the results of user testing the approach on a series of reviews.

Resubmission

26 Jun 2017 Submitted Manuscript version 2

9 Oct 2017 Author responded Author comments - Simon Lewin

General comments from the series editor	Author responses and changes made
Thanks for providing more methodological detail in the overview and subsequent papers. There are still some areas where it would be better if you could provide further details to reflect the amount of international developmental work undertaken e.g. databases searched, timeframes, how literature reviewed etc.	We have added further detail to the overall methods description in paper 1 of the series. Specifically, we have:  - Included the years during which we ran workshops and seminars to obtain feedback on CERQual, and the numbers of workshops and presentations undertaken - Specified the period during which small group feedback sessions were run - Specified the number of CERQual users and Project Group members interviewed In the component papers (papers 3-6), we have noted that the literature searches that we undertook were informal in nature, as follows (example from paper 5): "When developing CERQual's adequacy component, we undertook informal searches of the literature, including Google and Google Scholar, for definitions and discussion papers related to the concept of adequacy and to related concepts such as data quantity, sample size and data saturation." We have also elaborated on the methods used to develop the content of paper 7 – please see below.
Ethics statements. Papers state that no humans were involved. Suggest amending to reflect consensus approach, interviews and questionnaires undertaken.	As we did not undertake formal data collection with people – all data collection was informal, in the context of training workshops, presentations and assessments of use of the approach, we have changed the ethics approval and consent to participate statements to the following: "Not applicable. This study did not undertake any formal data collection involving humans or animals."
Titles and papers could reflect paper nth of # part in a series.	We have changed all titles to the following format, as agreed earlier (example from paper 1): 'Applying GRADE-CERQual to qualitative evidence synthesis findings – paper 1 of 7: Introduction to the series'

State of the art has been removed from paper 6 but not all of the other papers in the series.	'State of the art' has been removed from all papers in the series.
The new figure outlining the process is a good addition. As a reader I would have found it easier to read papers 3-6/7 before reading paper 2.	As discussed by email with Liz Glidewell, we had a very long debate within the group about this and concluded that there is no perfect order because paper 2 (overall assessment) and papers 3-6 (components) need to be seen together. We placed 'overall assessment' before the component papers as we felt that readers needed to understand what they were working towards before understanding each component. We feel that it would be best to keep the order as it is, but have made the following changes to assist readers: Papers 2, 3, 4, 5 and 6: We have inserted text along the lines of the following (example from paper 2 (p6): 'These component papers are closely related to this paper on making an overall CERQual assessment of confidence and creating a Summary of Qualitative Findings table. We have placed this paper before the four CERQual component papers as we think that it will be helpful for readers to understand how the component assessments will be used before discussing the details of how to apply each component.' Papers 2, 3, 4, 5 and 6: We have included in each paper an additional table that brings together all of the key definitions from each of the papers.
Do you still want to publish paper 7 as a standalone or incorporate it into the overview along with the other ongoing research?	Yes, we feel that it works best as a standalone paper.
Would the figure in the introduction outlining the process work better across all papers in the series as it contains more information than the figure just outlining the 4 and probable 5 th component?	Thanks for this very helpful suggestion which we have implemented across all of the papers.
1. Introduction	
The lack of such methods constrains the use of...suggest reframing to "methods may constrain".	Change made
"The CERQual approach is intended to be applied to well conducted syntheses." Could	We have not found this to be confusing in our interactions with users of CERQual. We feel that there would be little point in applying CERQual to a synthesis that has been poorly conducted as the

this be confusing to those applying the four components? Isn't CERQual designed to provide evidence of confidence in a well conducted syntheses?	findings of such a synthesis are unlikely to be reliable and the synthesis is unlikely report transparently the methods used or to include sufficient information on the primary studies to allow a CERQual assessment to be undertaken. We take the same approach in relation to GRADE for effectiveness, for the same reasons. The problem is sometimes colloquially called 'garbage in-garbage out'!
The section "Applying CERQual across types of qualitative data and syntheses methods". Would this be better placed after outlining how CERQual was developed?	We agree and have moved this section.
"supported other teams". Can you say any more about the scale or settings involved?	We have provided more detail as follows: "Thirdly, we applied the CERQual approach within diverse qualitative evidence syntheses in the areas of health and social care [6-8, 26-33] and also supported other teams in using CERQual by providing guidance through face-to-face or virtual training meetings and commenting on draft Summaries of Qualitative Findings tables. At least ten syntheses were supported in this way (for example, [34, 35])."
Can you provided further detail about the questionnaire and qualitative interviews?	We have now provided further detail in the text and added an additional file listing the questions covered. The revised text reads as follows: "We then gathered structured feedback from early users of CERQual through an online feedback form that was made available to all CERQual users and through short individual discussions with six members of review teams and two members of the CERQual Project Group. The questions included in the online feedback form and individual discussions are available in Additional File X."
Summarise important areas for methodological research from table 4 in text for the readers ease?	We have revised the text as follows: "Table 4 identifies several important areas for further methodological research, including how to apply CERQual in syntheses that include qualitative and quantitative data; how best to present CERQual assessments together with other kinds of evidence; ways of applying CERQual to syntheses of sources that have not used formal qualitative research procedures; and whether CERQual requires adaptation for application to more interpretive synthesis outputs, such as logic models."
2. Making an overall assessment and summary of qualitative findings	
Should the paragraph describing the four levels and rating down on p12 be moved to p10 under the 4 bulleted levels of concern?	This change has been made.

Place the text relating to variation in assessors after the text outlining who should undertake an assessment?	This change has been made.
Table 5. typo in component t missing. Should you advise assessors to report how they've handled variation in levels of concern?	This typo has been corrected.
3. Methodological limitations – problems design or conduct of primary studies	
Consider adding a brief description of the Evidence Profile to p12.	Ok. We have now added the following parentheses describing the evidence profile on page 12 following the sentence: “Where you have concerns about methodological limitations, describe these concerns in the CERQual Evidence Profile in sufficient detail to allow users of the review findings to understand the reasons for the assessments made (The Evidence Profile presents each review finding along with an explanation of its CERQual assessment)”
Link in text to table 2?	We have now added the following on page 9: “See Table 2 for an outline of areas where further work is needed with respect to critical appraisal tools for qualitative research.”
4. Coherence – How well finding supported by body of evidence 3500 3429	
Consider adding a brief description of the Evidence Profile to p13.	We have added a brief description of the evidence profile on page 12: “Where you have concerns about coherence, you should describe these concerns in the CERQual Evidence Profile in sufficient detail to allow users of the review findings to understand the reasons for the assessments made. The Evidence Profile presents each review finding along with the assessments for each CERQual component, the overall CERQual assessment for that finding and an explanation of this overall assessment. For more information, see the second paper in this series [19].”

5. Adequacy of data – degree of richness and quantity of data 3500 2507	
Consider contacting authors for further information as in other assessments?	We have added the following information to lines 204-205: “An overview of the number of studies from which this data originated, and where possible, the number of participants or observations. Information about the number of participants or observations supporting each finding may be difficult to gain from the individual studies. While most studies describe the number of participants they included in their study overall or give some indication of the extent of their observations, they may be less clear about how well represented participants are in different themes and categories. You can contact study authors for additional information, but they may not be able to readily provide this level of detail. In these cases, this lack of information should be noted, and your assessment of data adequacy will have to be made based on the information available.”
The sentence “For a description on descriptive and explanatory findings...” isn’t embedded.	We have moved this sentence to lines 232-233.
Consider adding a brief description of the Evidence Profile to p12.	We have added the following information to lines 277-279: The Evidence Profile presents each review finding along with the assessments for each CERQual component, the overall CERQual assessment for that finding and an explanation of this overall assessment.
6. Relevance – extent applicable to context (perspective or population, phenomenon of interest, setting) of review question 3500 3551	
I found a lot of the text more relevant to conducting a review than the CERQual assessment e.g. using theories and	Relevance is the only CERQual component that links directly to the review question. All the issues raised by the Editor need to be taken into consideration at the review design stage. We make this clear in the manuscript. See P6:

frameworks, how and when the review question should be developed, the pre-specification of sub-groups, strategies for identifying and selecting studies, trade-offs in searching.	'Relevance is the CERQual component that is anchored to the context specified in the review question. How the review question and objectives are expressed, how a priori subgroup analyses are specified, and how theoretical considerations inform the review design are therefore critical to making an assessment of relevance when applying CERQual.' See page 11: 'When assessing relevance, you should reflect on how the sample was located and on the underpinning principles that determined its selection....'
Word missing p13 "You should if possible, that this"	Sincere apologies, this typo was corrected previously but the corrected draft was not uploaded last time.
Is it possible to comment on how the levels of concern map onto the different threats to relevance 'partial', 'indirect' and 'unclear'?	Tables 3, 4, 5 and 6 provide visual examples. Sincere apologies, these tables may not have been uploaded in error last time.
7. Dissemination bias – selective dissemination of studies or findings 2000 2455	
Methodological details e.g. 'consulting relevant literature' and 'additional empirical work'	We have added further detail as follows: Abstract: "We developed this paper by gathering feedback from relevant research communities, searching MEDLINE and Web of Science to identify and characterize the existing literature discussing or assessing dissemination bias in qualitative research and its wider implications, developing consensus through project group meetings, and conducting an online survey of on the extent, awareness and perceptions of dissemination bias in qualitative research." Main text: "We used a pragmatic approach to develop our ideas on dissemination bias by consulting the literature on this topic, including searching MEDLINE and Web of Science to identify and characterize the existing literature discussing or assessing dissemination bias in qualitative research and its wider implications [3]; talking to experts in dissemination bias and qualitative evidence synthesis in a number of workshops; and developing consensus through multiple face-to-

		face CERQual Project Group meetings and teleconferences. We also undertook an online survey of researchers, journal editors and peer reviewers within the qualitative research domain on the extent, awareness and perceptions of dissemination bias in qualitative research [4].”
--	--	--

Resubmission 2

9 Oct 2017 Submitted Manuscript version 3

Publishing

17 Oct 2017 Editorially accepted

How does Open Peer Review work?

Open peer review is a system where authors know who the reviewers are, and the reviewers know who the authors are. If the manuscript is accepted, the named reviewer reports are published alongside the article. Pre-publication versions of the article and author comments to reviewers are available by contacting info@biomedcentral.com. All previous versions of the manuscript and all author responses to the reviewers are also available.

You can find further information about the peer review system here.